# Lipid metabolism and functional somatic disorders in the general population. The DanFunD study

Torben Jørgensen[1,2]*, Rikke Kart Jacobsen[1], Ditte Sæbye[1], Marie Weinreich Petersen[3,4], Per Fink[3,4], Lise Gormsen[3,4], Allan Linneberg[1,5], Anne Ahrendt Bjerregaard[1], Signe Ulfbeck Schovsbo[1], Michael Eriksen Benros[6], Lene Falgaard Eplov[6], Niklas Rye Jørgensen[5,7], Thomas Meinertz Dantoft[1]

1 Centre for Clinical Research and Prevention, Capital Region and University of Copenhagen, Copenhagen, Denmark, 2 Department of Public Health, Faculty of Health and Medical Science, University of Copenhagen, Copenhagen, Denmark, 3 Research Clinic for Functional Disorders and Psychosomatics, Aarhus University Hospital, Aarhus, Denmark, 4 Department of Clinical Medicine, Aarhus University, Aarhus, Denmark, 5 Department of Clinical Medicine, Faculty of Health and Medical Science, University of Copenhagen, Copenhagen, Denmark, 6 Copenhagen Research Center for Mental Health–CORE, Mental Health Centre Copenhagen, Copenhagen, Capital Region, Denmark, 7 Department of Clinical Biochemistry, Rigshospitalet, Copenhagen, Denmark

* torben.joergensen@regionh.dk

**Data Availability Statement:** Data cannot be made publicly available for ethical and legal reasons. Public availability may compromise participant privacy, and this would not comply with Danish

## Abstract

### Objectives

Earlier studies on the association between plasma lipid profiles and functional somatic disorders (FSD) are mainly small case control studies hampered by selection bias and do not consider the great overlap between the various FSDs. The aim of the present study was to investigate the associations between various FSDs and plasma lipid profiles (total cholesterol, HDL cholesterol, non-HDL cholesterol and triglycerides) in a large, unselected population.

### Design

A cross-sectional general population-based study

### Setting

The Danish Study of Functional Somatic Disorders (DanFunD) conducted in 2011–2015 in 10 municipalities in the western part of greater Copenhagen, Denmark.

### Participants

A total of 8,608 men and women aged 18–76 years were included in the analyses. Various delimitations of FSD such as chronic fatigue, chronic widespread pain, irritable bowel, and bodily distress syndrome were measured using validated self-administrated questionnaires. Lipid parameters were measured from fasting plasma samples using colorimetric slide methods with Vitros 4600/5600 Ortho Clinical Diagnostics.

legislation. Access to the subset of data included in this study can be gained through submitting a request to The Capital Region Knowledge Center for Data Compliance, The Capital Region Denmark; cru-fp-vfd@regionh.dk. Acquisition of data are only allowed after permission to handle data has been obtained in accordance with the guidelines stated by the Danish Data Protection Agency: http://www.datatilsynet.dk/english

**Funding:** Torben Jørgensen received financial support to the study from TrygFonden (7-11-0213) and the Lundbeck Foundation (R155-2013-14070). The sponsors did not play any role in the study design, data collection and analysis, decision to publish, oer preparation of the manuscript

**Competing interests:** The authors have declared that no competing interests exist.

## Outcome measures

Logistic regression analyses were used to calculate possible associations between plasma lipids and the various delimitations of FSD. Associations are presented by OR (95% CI) and shown in boxplots.

## Results

We found a positive association between bodily distress syndrome and triglycerides and non-HDL cholesterol and a negative association with HDL-cholesterol, but no consistent association with total cholesterol. A similar pattern was observed for persons with chronic fatigue, and to some degree for persons with chronic widespread pain, whereas persons with irritable bowel did not show a clear association with the lipid profiles.

## Conclusion

This is the first major study on plasma lipid profiles and FSD indicating an association between some delimitations of FSD and an unfavorable lipid profile. Due to the cross-sectional design, it cannot be determined whether the findings are consequences or determinants of FSD. Further studies–preferable prospective studies—are needed.

## Introduction

Functional Somatic Disorders (FSD) refer to clusters of recurrent disabling bodily symptoms (e.g. chronic fatigue, pain) and are prevalent in all medical settings [1]. FSD cannot be better explained by other somatic or psychiatric condition and varies from mild to severe symptoms that may cause extreme impairment [2]. FSD are associated with poor quality of life, and loss of labor market attachment [3, 4].

The literature comprises a huge variation of various delimitations of FSD, the most used are fibromyalgia (FM) or chronic widespread pain (CWP), chronic fatigue syndrome (CFS) and irritable bowel syndrome (IBS). A well-documented substantial overlap between these syndromes has led to the question whether they are distinct diseases or represent the same underlying condition [5, 6]. Therefore, the diagnostic construct of bodily distress syndrome (BDS) [7, 8] was introduced as a unifying diagnostic approach, a construct which has been verified in the general population [9]. The etiology of FSD is considered multifactorial, but as no clear physiological or biological markers has yet been identified [10], the diagnosis of the various FSD is based on identification of characteristic symptom patterns.

Some studies have shown an association between unfavorable plasma lipid profiles such as high levels of triglycerides and low density lipoprotein (LDL) cholesterol and low levels of high density lipoprotein (HDL) cholesterol in patients with FM [11–13], CFS [14, 15], and IBS [16], but most of these are smaller case-control studies in highly specialized clinical settings with diverse terminologies and high risk of selection bias. Also, the studies have only examined one delimitation of FSD at the time, ignoring the overlap that exists among the syndromes. Due to the many delimitations and their mutual overlap, it has been suggested to include more than just one delimitation when assessing associations with FSD [6].

Hypotheses dealing with a possible association between unfavorable lipid profiles and FSD include oxidative stress [14], presence of obesity [13], unfavorable lifestyle [16] and altered response to stress [15]. But the main problem in assessing any real association between FSD

and lipid profile is the lack of large-scale, population-based, epidemiological studies, where the various FSD are analyzed simultaneously. The objective of the present cross-sectional study was to investigate whether an unfavorable plasma lipid profiles was associated with various delimitations of FSD in a large population-based cohort, the Danish Study of Functional Disorders (DanFunD).

## Material and methods

### Study population and data collection

The DanFunD cohort has been described in detail elsewhere [17, 18]. In short, DanFunD is a population-based study comprising 9,656 (33.7% of the invited population) men and women aged 18–76 years and living in 10 municipalities in the Western part of greater Copenhagen, Denmark. The sample was randomly drawn from the Danish Civil Registration System. The participants should be born in Denmark, being Danish citizens, and not being pregnant. They were invited to the DanFunD study at the Research Centre for Prevention and Health in the period from 11/02/2011 to 06/08/2015 and examined between 11/10/2011 and 06/30/2015. All participants were asked to meet fasting at the day of the general health examination and to abstain from smoking at least 1 hour prior to examination. Analyses for the present manuscript was performed in the period 1/11.2022 to 31.03.2023. Among the 9,656 participants there were missing information on lipids or FSD in 85 and missing information on one or more covariate in 963, leaving 8,608 for analyses.

Written informed consent was obtained from each participant before participation, and the study was approved by the Ethical Committee of Copenhagen County (Ethics Committee: H-3-2011-081; H-3-2012-0015) and the Danish Data Protection Agency. Authors did not have access to information that could identify individual participants during or after data collection.

### Measurements

Due to the many delimitations of CFS, IBS, and CWP/FM we avoided the word syndrome and used instead chronic fatigue (CF), chronic widespread pain (CWP), and irritable bowel (IB). These conditions were identified with symptom lists from questionnaires, where only bothersome symptoms within the last 12 months were included [2]. Presence of IB symptoms and bothersome CWP symptoms were obtained on a four-point Likert scale from "never" to "almost constantly". Symptoms being present "frequently" or "almost constantly" were included. IB was assigned to participants having both abdominal pain and distension and, in addition, either borborygmi or altering stool consistency, or both [19]. CWP was assigned to participants having pain above and below the waist, in the right and the left sides of the body and in the axial skeleton [20]. Symptoms of CF were rated on a four-point Likert scale from "less than usual" to "much more than usual", and symptoms had to be present "more than usual" or "much more than usual" to be included. CF was assigned to participants scoring at least 4 on the 11-item fatigue scale [21]. Due to the known overlap between CF, CWP and IB, analyses were made both for each condition irrespectively of comorbid FSD (total) and the pure categories for each condition defined as the condition in question without the two other conditions.

Symptoms of BDS were rated on a five-point Likert scale ranging from "not at all" to "a lot" bothered by symptoms, and the categories "somewhat", "quite a bit", or "a lot" were included. BDS constitutes a single/oligo-organ type including persons with at least four symptoms from one or two of four symptom clusters and a multi-organ type (BDS-multi) comprising persons

with at least four symptoms from at least three of the four symptom clusters. In the analysis, BDS-total encompassed persons with either single/oligo or multi-BDS [22].

Total cholesterol, HDL cholesterol and triglycerides were measured from freshly collected fasting plasma samples using colorimetric slide methods with Vitros 4600/5600 Ortho Clinical Diagnostics (OCD). Non-HDL cholesterol was calculated by subtracting HDL cholesterol from total cholesterol.

## Covariates

Use of lipid lowering medicine was self-reported ("do you take lipid lowering medicine: yes/no") and will refer to use of statins. Physical activity was assessed by the physical activity scale (PAS 2) questionnaire adding up number of metabolic equivalents (METs) achieved during a common day [23]. Dietary intake was estimated using a self-administered 26-item food frequency questionnaire [FFQ]) classified in three categories (healthy, neutral, unhealthy) [24]. Alcohol was measured as number of drinks per week. Smoking status was assessed in four categories "daily smoker", "occasional smoker", "former smoker", or "never smoker". Subjective social class was measured as the position on a social 10-step staircase with 10 representing the highest social position [25]. Body mass index (BMI) was calculated from weight (with light indoor clothes) in kg divided by height (without shoes) in $m^2$ and divided into four categories ($< 20$; 20–24.99; 25–29,99; and 30+).

As regard comorbid conditions, the participants were asked "Have you ever been told by a physician that you suffer from . . .. . ..", for a number of conditions. Cardio-metabolic diseases were defined as former myocardial infarction, other heart disease, stroke, and diabetes and excluded in sensitivity analyses.

## Statistical analyses

The associations between plasma lipids and FSD were displayed in boxplots showing the mean, median, interquartile range and outliers of plasma lipids in the various FSD groups and were analyzed by a number of logistic regression analyses in two models with FSD as the dependent variables. In model 1, adjustment was made for age, sex, and lipid lowering medication; and in model 2, further adjustment was made for BMI, social position, and lifestyle factors (smoking, diet, physical activity, and alcohol intake).

Statistical analyses were performed using the logistic procedure in SAS Enterprise Guide version 7.15, (SAS Institute Inc., Cary, NC, USA).

Continuous variables were assessed for linearity with log odds of each FSD. Linearity was investigated by cubic splines with varying number of knots. Knots were placed at percentiles of the variable as suggested by Harrel [26]. Models were evaluated by graphical inspection, AIC and Likelihood Ratio tests. Linear splines were preferred for lipids as exposure. For each lipid interaction with sex and age were investigated in model 1 and if relevant included in both model 1 and 2. When interactions are included results/OR are presented for each sex or age group (for CF: above/below 50y, for CWP: <57y, 57-65y, >65y). Data are presented as OR with 95% confidence intervals (CI).

In sensitivity analyses the boxplots and analyses are repeated after excluding persons with known cardiometabolic diseases (N = 1033).

## Results

A total of 8,608 persons had information on all variables and could be included in the analyses (Table 1). The boxplots indicates that differences in lipid profiles between persons with FSD and no FSD are rather small (Figs 1, 2, and 3).

**Table 1. Distribution of independent variables and covariates according to sex among the 8,608 persons included in model 2.** The DanFunD study.

| Variable | Men | | Women | |
|---|---|---|---|---|
| | N | Median (25–75 percentile) | N | Median (25–75 percentile) |
| Age (years) | 4001 | 55 (45–64) | 4607 | 53 (44–63) |
| Total cholesterol (mmol/l) | 4000 | 5.20 (4.50–6.00) | 4605 | 5.30 (4.60–6.10) |
| Non-HDL cholesterol (mmol/l) | 4000 | 3.91 (3.23–4.66) | 4605 | 3.69 (3.04–4.42) |
| HDL cholesterol (mmol/l) | 4001 | 1.25 (1.05–1.50) | 4607 | 1.55 (1.31–1.86) |
| Triglycerides (mmol/l) | 4001 | 1.15 (0.83–1.69) | 4606 | 0.95 (0.72–1.34) |
| Body mass index (kg/m$^2$) | 4001 | 26.20 (24.00–28.80) | 4607 | 24.40 (22.00–27.70) |
| Alcoholic drinks/week | 4001 | 8.00 (3.50–15.00) | 4607 | 4.00 (1.00–7.00) |
| MET | 4001 | 39.50 (37.19–42.49) | 4607 | 39.04 (37.05–41.34) |
| Self-perceived social position | 4001 | 7.00 (6.00–8.00) | 4607 | 7.00 (6.00–8.00) |
| | | Proportion (%) | | Proportion (%) |
| Daily smoking | 4001 | 12.7 | 4607 | 12.4 |
| Unhealthy diet | 4001 | 14.5 | 4607 | 7.9 |
| Lipid lowering medicine | 4001 | 16.6 | 4607 | 11.4 |
| Body mass index (kg/m$^2$) | 4001 | | 4607 | |
| <20 | 67 | 1.7 | 382 | 8.3 |
| 20–24.99 | 1368 | 34,2 | 2148 | 46.6 |
| 25–29.99 | 1838 | 45.9 | 1387 | 30.1 |
| 30+ | 728 | 18.2 | 690 | 15.0 |

As regard total BDS the association with total cholesterol was not linear and only showed a significant positive association for persons with a cholesterol above 5.6 mmol/l. Non-HDL cholesterol was significantly positive associated with total BDS, whereas HDL cholesterol was significant negatively associated. The association to triglycerides were not linear and showed the strongest positive association in persons with triglycerides below 1.45 mmol/l. The same picture was seen for single BDS. For multi BDS there was no significant association with total cholesterol, but a significant negative association for HDL cholesterol and a significant positive association with triglycerides (Table 2). Results were only slightly attenuated when adjusting for BMI, lifestyle, and social factors (Table 2), which could not be performed for multi BDS due to too few cases. In analyses, where persons with cardiometabolic diseases were excluded, the general picture did not change (Table 1 and Fig 1).

CF was not significantly associated with total cholesterol (Table 3 and Fig 2).

Regarding non-HDL cholesterol a significant positive association with CF was seen only at higher concentrations of non-HDL cholesterol. A negative association with CF was seen with HDL cholesterol in the older age-group only, whereas CF was significantly positively associated with triglycerides. A slight attenuation was seen when adjusting for BMI, lifestyle, and social factors. The same trends were seen for pure CF (Table 3). In analyses where persons with cardio-metabolic diseases were excluded, the picture became clearer with a significant association with unfavorable lipid profile and CF pure (Fig 3 and Table 2).

IB was not significantly associated with total cholesterol, but a positive trend was seen for men, but not women. Regarding non-HDL cholesterol IB was significantly associated in men, but not in women (Table 4). IB was not significantly associated with HDL cholesterol, and for triglycerides there was not a linear association, but a trend towards a negative association at the lower concentrations of triglycerides and a positive association at higher levels. The same picture was seen when adjusting for BMI, lifestyle, and social factors. When looking at pure IB the picture attenuated somewhat, as pure IB was not associated with cholesterol, non-HDL

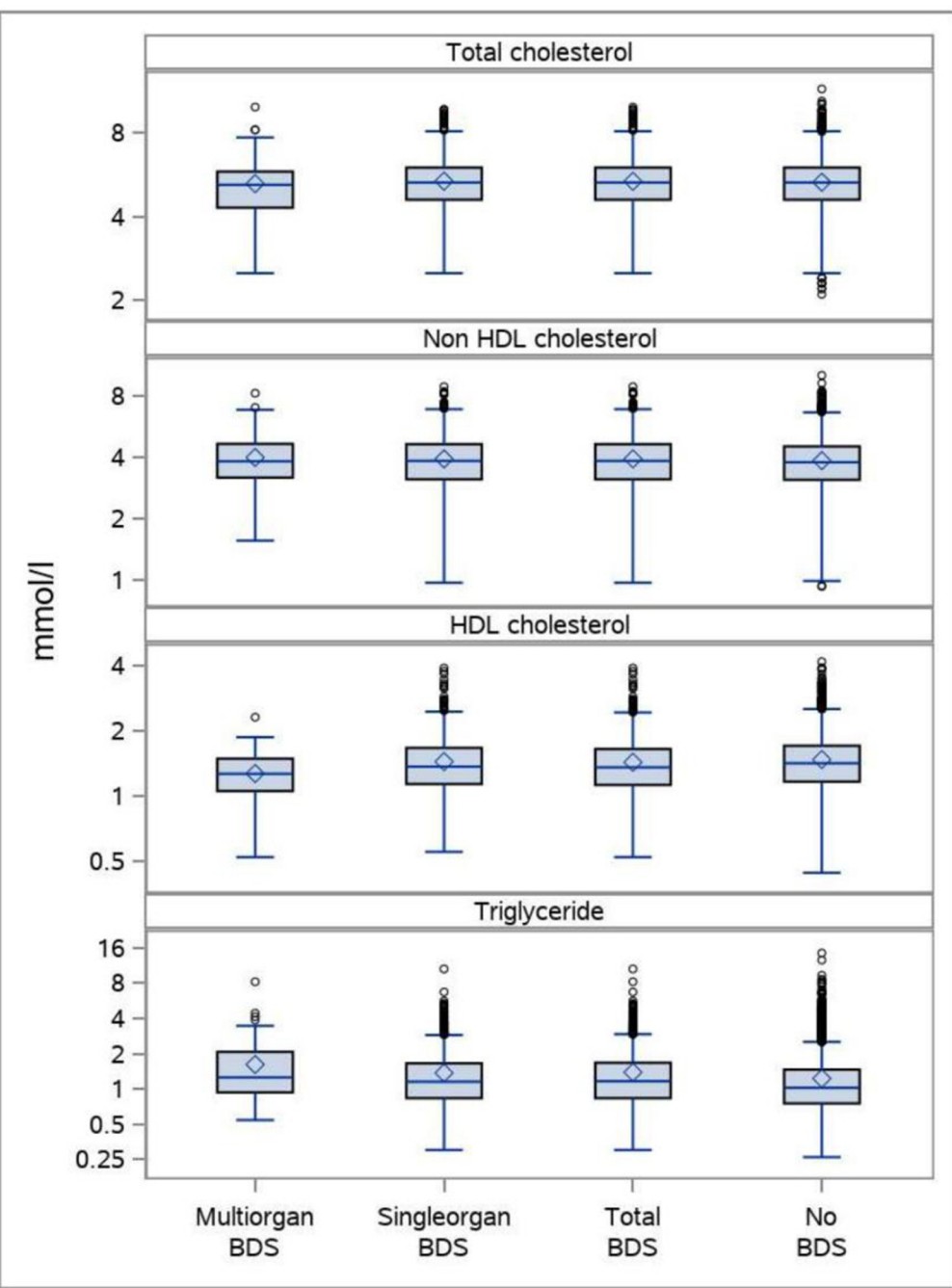

**Fig 1. The distribution of plasma lipids in persons with BDS (multi-organ BDS, single-organ BDS, and total BDS) and persons without BDS.** Boxplots showing median, mean, interquartile range, maximum observations below upper fence (1.5 IQR), minimum observations above lower fence (1.5 IQR), and maximum and minimum observations. The total DanFunD population.

cholesterol or HDL cholesterol, whereas there was a significant negative association at lower levels of triglycerides (Table 4). In analyses where persons with cardio-metabolic diseases were excluded the main picture was a negative association among women to non-HDL cholesterol and a negative association in the lower levels of triglycerides (Table 3).

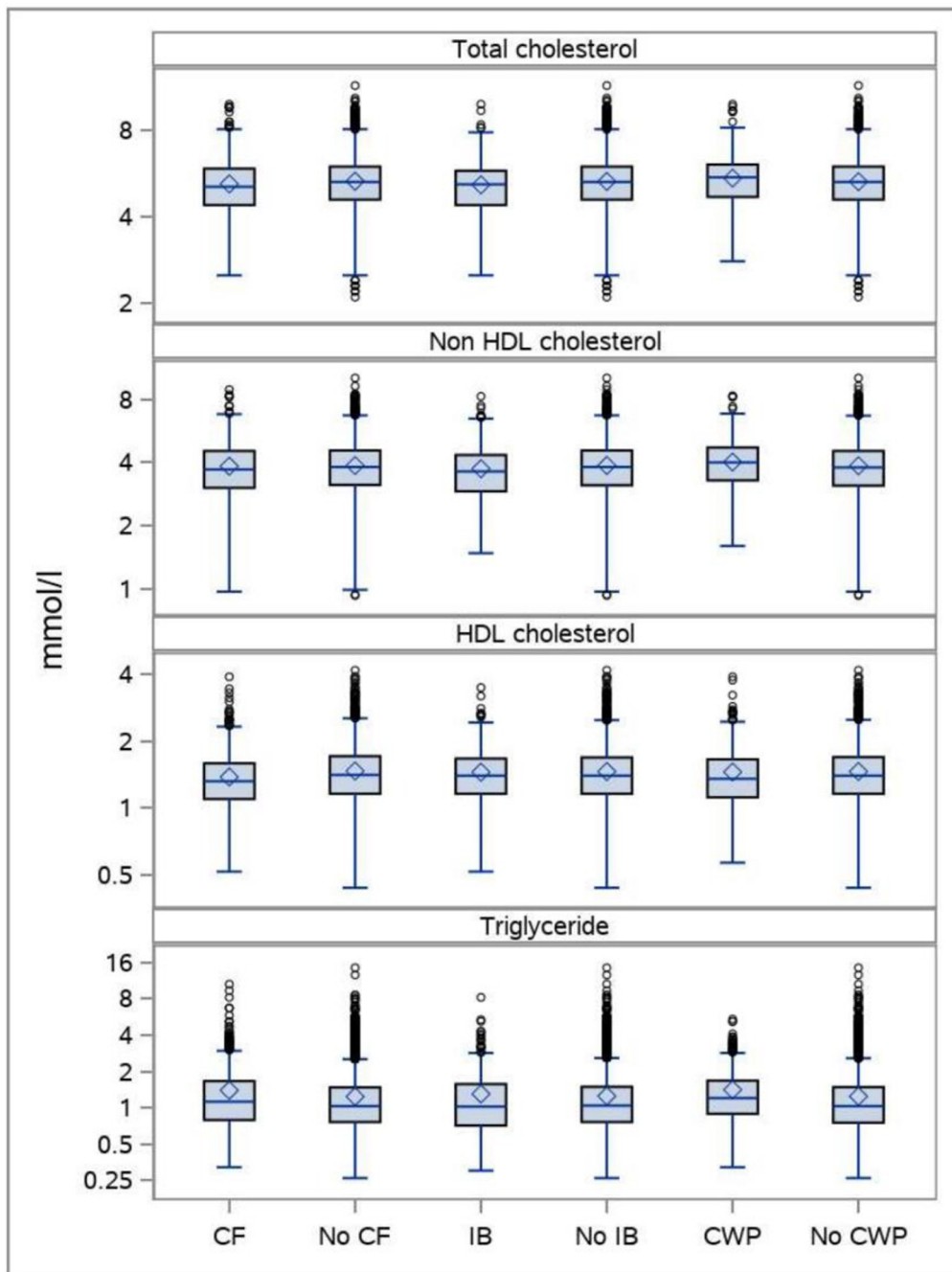

**Fig 2. The distribution of plasma lipids in persons with CF, no CF, IB, no IB, CWP, and no CWP.** Boxplots showing median, mean, interquartile range, maximum observations below upper fence (1.5 IQR), minimum observations above lower fence (1.5 IQR), and maximum and minimum observations. The total DanFunD population.

CWP was not significantly associated with total cholesterol, but positively associated with non-HDL cholesterol, however, the association disappeared when adjusting for lifestyle, BMI, and social factors (Table 5). Association with HDL cholesterol was not linear and showed diverging results. As regard triglycerides, a positive association was only found in the middle-aged group. For pure CWP there was a positive association to total cholesterol and non-HDL

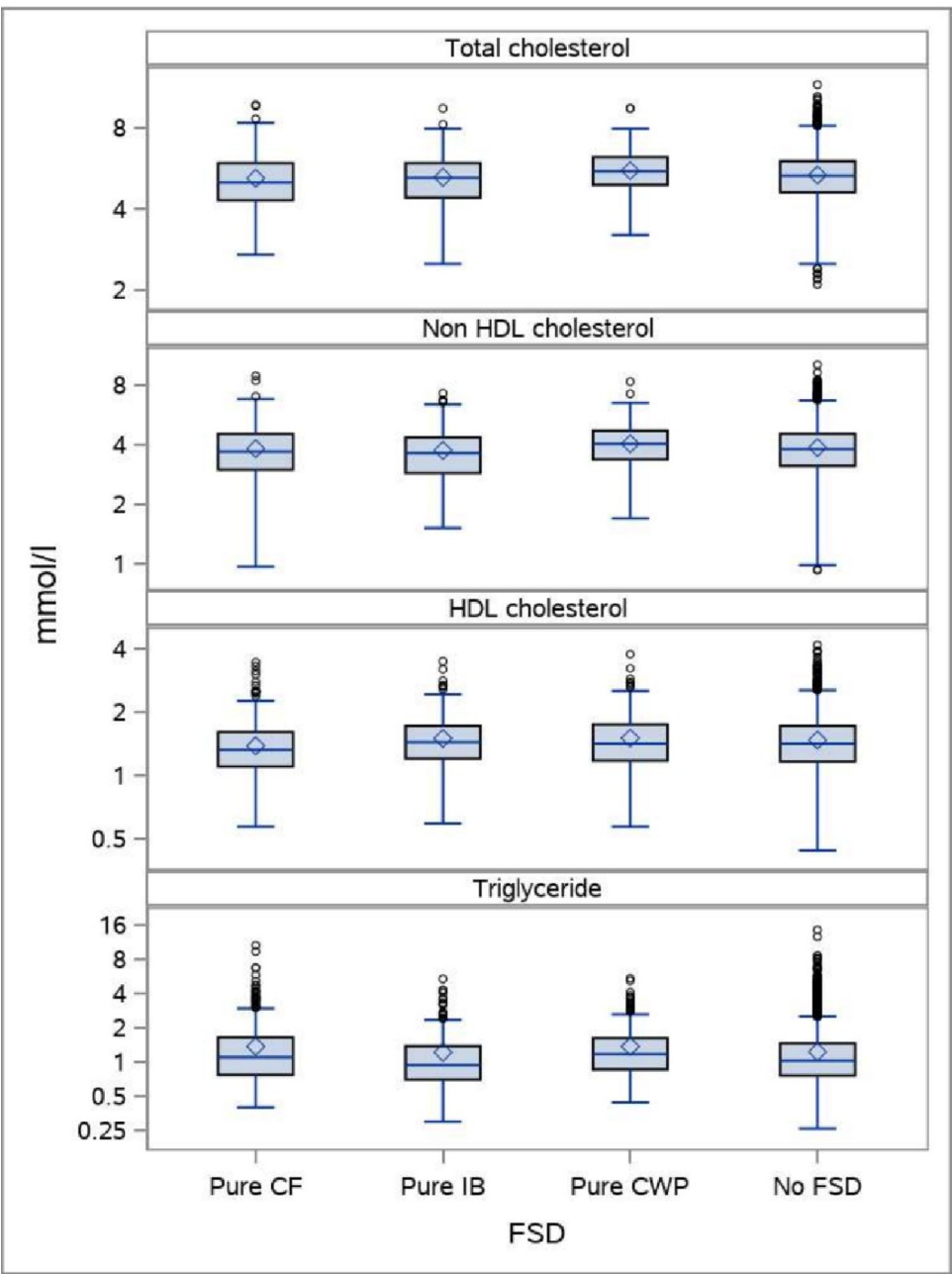

**Fig 3. The distribution of plasma lipids in persons with pure CF, pure IB, pure CWP, and neither CF, IB, nor CWP.** Boxplots showing median, mean, interquartile range, maximum observations below upper fence (1.5 IQR), minimum observations above lower fence (1.5 IQR), and maximum and minimum observations. The total DanFunD cohort.

cholesterol in the younger age group only. For HDL-cholesterol the picture was similar to total CWP. As regard triglycerides there was a small positive association (Table 5) with pure CWP. In analyses excluding persons with cardio-metabolic diseases the picture did not change substantial (Table 4).

**Table 2. Bodily Distress Syndrome (BDS) and plasma lipids in the total DanFunD cohort.** Odds ratio (95% confidence intervals) for a 0.1 mmol/l change in lipids.

| | Model 1 | Model 2 |
|---|---|---|
| **Total BDS* >< no BDS** | | |
| *Total cholesterol (N: 1297 ><7288)* | | |
| <5.6 mmol/l | 0.997 (0.987–1.009) | 0.996 (0.985–1.007) |
| > = 5.6 mmol/l | *1.016 (1.004–1.027)* | *1.012 (1.001–1.024)* |
| *Non-HDL cholesterol (N: 1297><7288)* | | |
| | *1.016 (1.010–1.022)* | *1.009 (1.002–1.015)* |
| *HDL cholesterol (N: 1300><7288)* | | |
| | *0.938 (0.923–0.953)* | *0.967 (0.951–0.984)* |
| *Triglycerides (N: 1299><7288)* | | |
| <1.45 mmol/l | *1.083 (1.061–1.106)* | *1.047 (1.024–1.070)* |
| > = 1.45 mmol/l | *1.011 (1.001–1.021)* | 1.005 (0.995–1.015) |
| **Single BDS >< no BDS ()** | | |
| *Total cholesterol (N: 1225><7288)* | | |
| <5.6 mmol/l | 0.999 (0.988–1.010) | 0.997 (0.986–1.009) |
| > = 5.6 mmol/l | *1.014 (1.002–1.025)* | 1.011 (0.999–1.023) |
| *Non-HDL cholesterol (N: 1225><7288)* | | |
| | *1.015 (1.009–1.022)* | *1.008 (1.002–1.015)* |
| *HDL cholesterol (N: 1227><7288)* | | |
| | *0.944 (0.928–0.959)* | *0.972 (0.955–0.989)* |
| *Triglycerides (N: 1226><7288)* | | |
| <1.45 mmol/l | *1.083 (1.060–1.106)* | *1.048 (1.024–1.072)* |
| > = 1.45 mmol/l | 1.008 (0.998–1.018) | 1.003 (0.992–1.014) |
| **Multi BDS >< no BDS** | | |
| *Total cholesterol (N:71><7288)* | | |
| <5.6 mmol/l | 0.980 (0.942–1.022) | Too few cases |
| > = 5.6 mmol/l | 1.034 (0.991–1.072) | - |
| *Non-HDL cholesterol (N: 71><7288)* | | |
| <3 mmol/l | 1.100 (0.979–1.275) | - |
| 3–4.54 mmol/l | 0.993 (0.943–1.046) | - |
| > = 4.55 mmol/l | *1.058 (1.010–1.098)* | - |
| *HDL cholesterol (N: 72><7288)* | | |
| | *0.819 (0.759–0.880)* | - |
| *Triglycerides (N: 72><7288)* | | |
| | *1.045 (1.027–1.062)* | - |

Model 1: adjusted for age, sex and lipid lowering drugs; Model 2: Further adjusted for body mass index, lifestyle (smoking, alcohol intake, physical activity, dietary habits), and social factors. One participant with BDS is not included in neither multi nor single organ BDS. Numbers in bold and italic indicate significant odds ratios.

## Discussion

In this first large epidemiological study with specific focus on FSD and lipid profile we found a small, but positive association between BDS and triglycerides and non-HDL cholesterol and a negative association with HDL-cholesterol. However, no consistent association with total cholesterol was found. The same pattern was seen for CF, and to some degree for CWP, whereas IB did not show a clear association with unfavorable plasma lipid profile.

**Table 3. Chronic fatigue (CF) and plasma lipids in the total DanFunD cohort.** Odds ratio (95% confidence intervals) for a 0.1 mmol/l change in lipids.

| | Model 1 | | Model 2 | |
|---|---|---|---|---|
| **CF >< no CF** | | | | |
| *Total cholesterol (N: 679><7862)* | | | | |
| | Age[a]<50 year | Age> = 50 years | Age<50 years | Age> = 50 years |
| <5.6 mmol/l | 1.012 (0.990–1.036) | 0.981 (0.949–1.015) | 1.009 (0.986–1.034) | 0.992 (0.959–1.028) |
| > = 5.6 mmol/l | 1.050 (0.992–1.107) | 0.992 (0.955–1.028) | 1.048 (0.987–1.108) | 0.990 (0.953–1.026) |
| *Non-HDL cholesterol (N: 679><7862)* | | | | |
| <3 mmol/l | 1.028 (0.993–1.066) | | 1.025 (0.989–1.064) | |
| 3–4.54 mmol/l | 1.011 (0.993–1.029) | | 0.998 (0.980–1.017) | |
| > = 4.55 mmol/l | *1.026 (1.007–1.045)* | | *1.021 (1.001–1.039)* | |
| *HDL cholesterol (N: 679><7865)* | | | | |
| | Age[b]<50 years | Age> = 50 years | Age<50 years | Age> = 50 years |
| | 0.966 (0.925–1.007) | *0.873 (0.828–0.919)* | 1.001 (0.958–1.044) | *0.904 (0.857–0.952)* |
| *Triglycerides (N: 679><7864)* | | | | |
| <1.45 mmol/l | *1.076 (1.048–1.106)* | | *1.040 (1.010–1.071)* | |
| > = 1.45 mmol/l | *1.024 (1.012–1.035)* | | *1.018 (1.006–1.030)* | |
| **Pure CF >< no CF, IB or CWP** | | | | |
| *Total cholesterol (N: 480><7326)* | | | | |
| | Age[c]<50 years | Age> = 50 years | Age<50 years | Age> = 50 years |
| <5.6 mmol/l | 1.022 (0.996–1.050) | 0.971 (0.934–1.012) | 1.017 (0.990–1.046) | 0.978 (0.940–1.020) |
| > = 5.6 mmol/l | 1.048 (0.983–1.111) | 0.981 (0.933–1.027) | 1.047 (0.980–1.113) | 0.979 (0.931–1.024) |
| *Non-HDL cholesterol (N: 480><7326)* | | | | |
| | *1.016 (1.006–1.026)* | | 1.009 (0.998–1.019) | |
| *HDL cholesterol (N: 480><7328)* | | | | |
| | Age[d]<50 years | Age> = 50 years | Age<50 years | Age > = 50 years |
| | 0.977 (0.930–1.025) | *0.865 (0.809–0.922)* | 1.002 (0.953–1.052) | *0.888 (0.830–0.947)* |
| *Triglycerides (N: 480><7327)* | | | | |
| | *1.032 (1.022–1.042)* | | *1.022 (1.011–1.032)* | |

Model 1: adjusted for age, sex and lipid lowering drugs; Model 2: Further adjusted for body mass index, lifestyle (smoking, alcohol intake, physical activity, dietary habits), and social factors. Numbers in bold and italic indicate significant odds ratios.

[a] Test for interaction: p = 0.0081.

[b] Test for interaction: p = 0.0100.

[c] Test for interaction: p = 0.0004.

[d] Test for interaction: p = 0.0099.

The literature gives a mixed picture of the association between FSD and plasma lipid profiles. One small case-control study [27] found a positive association between total cholesterol and CFS, whereas another case-control study did not [14]. The latter study, on the other hand, did see a negative association to HDL cholesterol. Two small case-control studies found positive association between CFS and triglycerides [14, 28], and one population-based case-control study [15] found a positive association between CFS and the metabolic syndrome, which includes high triglycerides and low HDL cholesterol. As regards CWP or FM, two case-control studies found a positive association to total cholesterol and LDL cholesterol [11, 12], whereas a third did not [29]. Two studies did not find an association between CWP/FM and HDL cholesterol [12, 29]. Two studies found a positive association between CWP/FM and triglycerides [11, 29], whereas a third did not [12]. A small case-control study did not find an association between IBS and cholesterol [30], whereas a Japanese cross-sectional population-based study

**Table 4. Irritable bowel (IB) and plasma lipids in the total DanFunD cohort.** Odds ratio (95% confidence intervals) for a 0.1 mmol/l change in lipids.

| | Model 1 | | Model 2 | |
|---|---|---|---|---|
| **IB >< no** | | | | |
| *Total cholesterol (N: 290><8183)* | | | | |
| | Men[a] | Women | Men | Women |
| | 1.023 (1.000–1.045) | 0.988 (0.973–1.002) | 1.020 (0.997–1.043) | 0.986 (0.971–1.000) |
| *Non-HDL cholesterol (N: 290><8183)* | | | | |
| | Men[b] | Women | Men | Women |
| | *1.027 (1.004–1.049)* | 0.993 (0.978–1.007) | *1.022 (1.000–1.045)* | 0.988 (0.973–1.003) |
| *HDL cholesterol (N: 291><8185)* | | | | |
| | 0.971 (0.941–1.002) | | 0.983 (0.950–1.016) | |
| *Triglycerides (N: 291><8184)* | | | | |
| <1.0 mmol/l | 0.923 (0.848–1.004) | | *0.903 (0.829–0.984)* | |
| 1.0–1.79 mmol/l | *1.069 (1.016–1.124)* | | *1.054 (1.001–1.111)* | |
| > = 1.8 mmol/l | 1.018 (0.996–1.035) | | 1.015 (0.992–1.033) | |
| **Pure IB >< no IB, CF or CWP** | | | | |
| *Total cholesterol (N: 176><7326)* | | | | |
| | Men[c] | Women | Men | Women |
| | 1.020 (0.994–1.047) | 0.983 (0.963–1.002) | 1.017 (0.991–1.044) | 0.982 (0.962–1.001) |
| *Non-HDL cholesterol (N: 176><7326)* | | | | |
| | Men[d] | Women | Men | Women |
| | 1.024 (0.998–1.049) | 0.982 (0.962–1.002) | 1.021 (0.995–1.047) | 0.980 (0.960–1.000) |
| *HDL cholesterol (N: 177><7328)* | | | | |
| | 0.995 (0.957–1.034) | | 0.996 (0.955–1.038) | |
| *Triglycerides (N: 177><7327)* | | | | |
| <1.0 mmol/l | *0.890 (0.803–0.988)* | | *0.880 (0.792–0.979)* | |
| 1.0–1.79 mmol/l | 1.054 (0.986–1.126) | | 1.045 (0.976–1.119) | |
| > = 1.8 mmol/l | 1.015 (0.981–1.038) | | 1.015 (0.980–1.038) | |

Model 1: adjusted for age, sex and lipid lowering drugs; Model 2: Further adjusted for body mass index, lifestyle (smoking, alcohol intake, physical activity, dietary habits), and social factors. Numbers in bold and italic indicate significant odds ratios.

[a] Test for interaction: p = 0.0074.

[b] Test for interaction: p = 0.0098.

[c] Test for interaction: p = 0.0189.

[d] Test for interaction: p = 0.0092.

[16] did find a positive association between IBS and triglycerides, but not with HDL or LDL cholesterol. There is no literature on a possible association between BDS and plasma lipids. An earlier study in the DanFunD cohort showed no association between plasma lipids and another delimitation of FSD (multiple chemical sensitivity) [31]. A systematic review of a variety of predictors of persistent somatic symptoms in the general population [32] identified cohort studies looking at plasma lipid and development of FSD with mixed results. The identified studies did not have lipid metabolism and FSD as focus but were mostly dealing with data from national registries identifying ICD codes on hyperlipidemia and FSD. Another cohort study [33] showed that a lifetime diagnosis of high cholesterol was associated with development of self-reported CFS.

These findings illustrate the need for studies with specific focus on FSD and they do not contradict our findings, which indicate an association between some of the FSD delimitations and an unfavorable plasma lipid profile. It is noteworthy that especially triglycerides seem to

**Table 5. Chronic widespread pain (CWP) and plasma lipids in the total DanFunD cohort.** Odds ratio (95% confidence intervals) for a 0.1 mmol/l change in lipids.

| | Model 1 | | | Model 2 | | |
|---|---|---|---|---|---|---|
| **CWP >< no CWP** | | | | | | |
| *Total cholesterol (N: 362><8176)* | | | | | | |
| | 1.006 (0.994–1.017) | | | 1.004 (0.993–1.015) | | |
| *Non-HDL cholesterol (N: 362><8176)* | | | | | | |
| | *1.016 (1.005–1.027)* | | | 1.006 (0.995–1.017) | | |
| *HDL cholesterol (N: 362><8179)* | | | | | | |
| <1.18 mmol/l | 0.954 (0.846–1.084) | | | 1.017 (0.898–1.159) | | |
| 1.18–1.79 mmol/l | *0.856 (0.805–0.909)* | | | *0.912 (0.857–0.971)* | | |
| > = 1.8 mmol/l | 1.054 (0.993–1.113) | | | *1.079 (1.016–1.140)* | | |
| *Triglycerides (N: 362><8178)* | | | | | | |
| | <57[a] | 57–65 | >65 | < 57 | 57–65 | >65 |
| <0.9 mmol/l | 1.072 (0.812–1.476) | *1.451 (1.224–1.742)* | 0.994 (0.780–1.298) | 1.005 (0.754–1.395) | *1.295 (1.090–1.558)* | 0.887 (0.692–1.165) |
| >0.9 mmol/l | 0.984 (0.896–1.051) | *1.022 (1.002–1.040)* | 1.032 (0.992–1.067) | 0.961 (0.871–1.035) | 1.009 (0.986–1.029) | 1.021 (0.979–1.060) |
| **Pure CWP >< no CWP, IB or CF** | | | | | | |
| *Total cholesterol (N: 214><7326)* | | | | | | |
| | <57[b] | 57–65 | >65 | <57 | 57–65 | >65 |
| | *1.104 (1.027–1.186)* | 0.991 (0.972–1.011) | 0.978 (0.951–1.006) | *1.089 (1.013–1.171)* | 0.992 (0.973–1.012) | 0.980 (0.952–1.008) |
| *Non-HDL cholesterol (N: 214><7326)* | | | | | | |
| | <57[c] | 57–65 | >65 | <57 | 57–65 | >65 |
| | *1.101 (1.028–1.176)* | 1.002 (0.983–1.020) | 0.985 (0.957–1.013) | *1.085 (1.012–1.161)* | 0.995 (0.976–1.014) | 0.977 (0.949–1.006) |
| *HDL cholesterol (N: 214><7328)* | | | | | | |
| <1.18 mmol/l | 1.005 (0.851–1.206) | | | 1.049 (0.885–1.263) | | |
| 1.18–1.79 mmol/l | *0.862 (0.798–0.931)* | | | *0.908 (0.838–0.983)* | | |
| > = 1.8 mmol/l | 1.070 (0.994–1.142) | | | *1.095 (1.017–1.169)* | | |
| *Triglycerides (N: 214><7327)* | | | | | | |
| <0.9 mmol/l | *1.210 (1.051–1.411)* | | | 1.119 (0.969–1.308) | | |
| > = 0.9 mmol/l | 1.015 (0.995–1.032) | | | 1.004 (0.981–1.023) | | |

Model 1: adjusted for age, sex and lipid lowering drugs; Model 2: Further adjusted for body mass index, lifestyle (smoking, alcohol intake, physical activity, dietary habits), and social factors. Numbers in bold and italic indicate significant odds ratios.

[a] Test for interaction: p = 0.0294.

[b] Test for interaction: p = 0.0124.

[c] Test for interaction: p = 0.0220.

be positively associated with most delimitations. The lack of a clear association between total cholesterol and FSDs could be expected, as total cholesterol is made up of both "healthy" and "unhealthy" cholesterol, which also stresses that it is necessary to go into details with the lipid parameters. In this study we use non-HDL cholesterol as a measure of unhealthy cholesterol instead of using both LDL cholesterol and VLDL cholesterol, where the latter is firmly tied with the triglyceride levels.

Possible explanations for the findings could be the well-known unfavorable lifestyle and obesity among persons with FSD [15, 34] but adjusting for these parameters in our analyses did not change the findings substantially. As chronic stress is suggested to be one of the mechanisms behind FSD, a possible explanation of the findings in this study could be a response to

chronic stress, which leads to alteration of the hypothalamic-pituitary-adrenal (HPA) axis [35, 36].The HPA-axis is central to the construct of allostatic load, a physiological measure of the cumulative wear and tear of the body due to repeated cycles of attempts to adapt to changes. Physiological effects of allostatic load can lead to a dysregulated lipid metabolism. Prospective studies are needed to further explore this area.

The different results between CF, CWP, and IB (especially as regard the pure delimitations) could indicate different disease mechanisms in the various delimitations of FSD, which add to the discussion whether we are talking about one or several diseases.

## Strengths and limitations

The current study has several strengths: First, we include a large (N = 9,656) unselected sample from the general adult population with almost equal distribution of the two sexes. Other studies mostly involve highly selected–often female—patient samples recruited in specialized clinical setting with great risk of publication bias. The population-based study design reduces the risk of selection bias and allows the results to be generalized to other adult populations. Second, as many different criteria to identify FSD have been proposed, we included two approaches for defining FSD in our study. Hence, we tried to capture the diverse nature of these conditions as both mono- and multi-systematic. Third, we used well-known and validated symptom questionnaires for defining the various FSD.

However, our study also has some limitations: First, it may be that we included cases with milder symptoms than studies in patient cohorts. However, in our study, a cut-off on symptom severity was made, only including bothering symptoms in the criteria defining FSD cases. Furthermore, we included delimitation of multi-systematic conditions (multi-organ BDS). We therefore argue that the cases in our study were not all mild cases but also constituted individuals with symptom patterns of severe FSD. Second, the response rate of 29.5% may be considered low, and even though the risk of selection bias is markedly reduced compared to clinical studies, we cannot rule it completely out. But we have shown that selection bias is not a major problem in the DanFunD study [37]. Third, we are dealing with a cross-sectional design, and therefore cannot determine whether the findings are consequences or determinants for FSD. Finally, as the case definitions is based on questionaries, we cannot rule out that some of the cases are misclassified but excluding persons with cardiometabolic diseases did not change the results.

## Conclusions

In this first large population-based study about FSD and lipid profile, we found a small, but significant association between various delimitations of FSD and an unfavorable lipid profile. The findings were clear for the unifying bodily distress syndrome BDS and for CF, but more uncertain as regards CWP, whereas IB did not show this association. Further studies–preferable prospective studies—are needed to explore these findings.

## Acknowledgments

We want to acknowledge the staff of the DanFunD study for their huge work. We also want to thank the many participants for using spare time to contribute with data for the study.

## Author Contributions

**Conceptualization:** Torben Jørgensen, Lene Falgaard Eplov, Thomas Meinertz Dantoft.

**Data curation:** Torben Jørgensen, Niklas Rye Jørgensen.

**Formal analysis:** Torben Jørgensen, Rikke Kart Jacobsen, Ditte Sæbye.

**Funding acquisition:** Torben Jørgensen, Per Fink.

**Investigation:** Torben Jørgensen.

**Methodology:** Torben Jørgensen, Marie Weinreich Petersen, Per Fink, Lise Gormsen, Allan Linneberg, Anne Ahrendt Bjerregaard, Michael Eriksen Benros, Lene Falgaard Eplov, Niklas Rye Jørgensen, Thomas Meinertz Dantoft.

**Project administration:** Torben Jørgensen.

**Resources:** Torben Jørgensen.

**Software:** Torben Jørgensen.

**Supervision:** Torben Jørgensen.

**Validation:** Torben Jørgensen.

**Visualization:** Torben Jørgensen.

**Writing – original draft:** Torben Jørgensen.

**Writing – review & editing:** Torben Jørgensen, Rikke Kart Jacobsen, Ditte Sæbye, Marie Weinreich Petersen, Per Fink, Lise Gormsen, Allan Linneberg, Anne Ahrendt Bjerregaard, Signe Ulfbeck Schovsbo, Michael Eriksen Benros, Lene Falgaard Eplov, Niklas Rye Jørgensen, Thomas Meinertz Dantoft.

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
