## [Decision Letter · Decision Letter 0]

5 Oct 2023

PONE-D-23-13283Lipid metabolism and functional somatic disorders in the general population. The DanFunD studyPLOS ONE

Dear Dr. Jørgensen,

Thank you for submitting your manuscript to PLOS ONE. After careful consideration, we feel that it has merit but does not fully meet PLOS ONE’s publication criteria as it currently stands. Therefore, we invite you to submit a revised version of the manuscript that addresses the points raised during the review process.

We look forward to receiving your revised manuscript.

Kind regards,

Aleksandra Klisic

Academic Editor

PLOS ONE

Journal Requirements:

2. Thank you for stating the following in the Acknowledgments "We want to acknowledge the staff of the DanFunD study for their huge work. We also want to thank the many participant for using spare time to contribute with data for the study. The study was financially supported by TrygFonden (7-11-0213) and the Lundbeck Foundation (R155-2013-14070). The funding sources had no influence on study design, data collection, data-analyses and results."

"Torben Jørgensen received financial support to the study from TrygFonden (7-11-0213) and the Lundbeck Foundation (R155-2013-14070). The sponsors did not play any role in the study design, data collection and analysis, decision to publish, oer preparation of the manuscript"

5. Please note that supplementary tables (should remain/ be uploaded) as separate "supporting information" files

Reviewers' comments:

Reviewer's Responses to Questions

**Comments to the Author**

1. Is the manuscript technically sound, and do the data support the conclusions?

Reviewer #1: Partly

Reviewer #2: Yes

Reviewer #3: Yes

2. Has the statistical analysis been performed appropriately and rigorously? 

Reviewer #1: No

Reviewer #2: Yes

Reviewer #3: Yes

3. Have the authors made all data underlying the findings in their manuscript fully available?

Reviewer #1: Yes

Reviewer #2: Yes

Reviewer #3: Yes

4. Is the manuscript presented in an intelligible fashion and written in standard English?

Reviewer #1: Yes

Reviewer #2: Yes

Reviewer #3: Yes

5. Review Comments to the Author

Reviewer #1: This paper presents the results of primary research which have not, to my knowledge, been published previously. It has a number of strengths including a large, population-based sample and objective outcome measures. The research meets high ethical standards.

There a number of problems with the work and its presentation. These are:

1) Although the sample is truly population-based the low response rate (30%) raises the possibility of a biased sample. Are there other publications from DanFunD that indicate it is representative of the local population?

2) The introduction (line 87) refers to “the lack of large-scale, population-based, epidemiological studies, where the various FSD are analyzed simultaneously (17). This is a misquote. Reference 17 presents the “case for longitudinal population-based studies in the field of functional somatic syndromes.” This is a serious shortcoming of the present study which is cross-sectional, although the authors do admit (line 358) “ we are dealing with a cross-sectional design, and therefore cannot determine whether the findings are consequences or determinants for FSD.

3) The literature review is rather limited. A recent paper (see A below) found 13 studies that had examined lipid profiles as predictors of FSS. That paper quoted evidence that a raised BMI predicted FM whereas the opposite was true in IBS. Another prospective population-based study also reported this finding (B below).

4) The theoretical basis for the whole project is not very clear. The stated aim is “to investigate possible associations between plasma lipid profiles and Functional Somatic disorders in a large population sample”. But the functional somatic disorders represents a group of disorders and there is no theoretical reason why all the different disorders (BDS single, Multi BDS, CF, IB, CWP -pure types and with other syndromes) should all have the same relationship with lipid profiles. Why should single BDS , which may be cardiopulmonary, gastrointestinal, musculoskeletal or general symptoms type share a common relationship with lipid profiles? The theoretical basis should be explained more fully.

5) This lack of clarity has led to multiple statistical tests. The figures indicate that 112 tests were performed and there seems to be little theoretical rational that can guide the reader through such a myriad of results. The outcome measure consists of 4 measures (total cholesterol, non-HDL cholesterol, HDL-cholesterol, triglycerides). These 4 measures are compared in each of 5 diagnostic categories with syndrome-free participants but each of the diagnostic categories are further split into pure and other types. This muddies the waters a lot. The results have then been adjusted for BMI, lifestyle, and social factors and some analysis are repeated in the sexes separately and by age groups. A clearer statistical plan could, surely, reduce the number of test results.

6) The very large number of tests means that no clear conclusion can be reached. The statement that “an association between some delimitations of FSD and an unfavorable lipid profile” is not satisfactory. In fact, the results are compatible with interpretation that BDS is a group of different disorders that have little in common. Not only do the characteristic symptoms differ in CF, IB and CWP, they appear to have a different relationship with lipid levels. The most likely explanation is that some of the subgroups of CFS have a raised BMI which may concur with the present results (D). Overall the results suggest that the notion of lumping together different disorders into a single group makes little sense when it comes to assessing correlates. It is known that BMI does not predict numerous somatic symptoms (the hallmark of Multi BDS) see C

A Kitselaar M, van der Vaart R et al. Predictors of Persistent Somatic Symptoms in the General Population: A Systematic Review of Cohort Studies Psychosomatic Medicine 2023 Jan 1;85(1):71-78.

B Monden, R, Rosmalen, JGM et al Predictors of new onsets of irritable bowel syndrome, chronic fatigue syndrome and fibromyalgia: the lifelines study. Psychol Med. 2022 Jan;52(1):112-120.

C Creed F. The Predictors of Somatic Symptoms in a Population Sample: The Lifelines Cohort Study. Psychosom Med 2022 Nov-Dec;84(9):1056-1066.

D Vollmer-Conna U, Aslakson E, White PD. An empirical delineation of the heterogeneity of chronic unexplained fatigue in women. Pharmacogenomics 2006 Apr;7(3):355-64.

Minor points

7) Abstract Line 44. This should read .”.. validated self – administered questionnaires if this was the case.

8) Spelling mistakes eg. (lines 186, 190) Tabel 1 and 2 In table 2 “To few cases” (should be Too few..)

9) Figures add little because so many similar boxes make it impossible to spot differences where these occur.

Reviewer #2: The manuscript is very interesting, addressing a topic of interest, while it is well-written and structured.

Some minor comments:

1. I think that the definition of cardio-metabolic diseases should also include MAFLD, CKD, etc., besides atherosclerotic CVD and diabetes mellitus.

2. Information regarding the type of lipid-lowering treatment (statin vs. non-statin) would be useful.

3. Of note, did the researchers have any available data concerning the association of IBS with lipid parameters, according to type of IBS (IBS-D, IBS-C, IBS-M), since, for example, subjects with IBS-D may experience more strict dietary pattern than IBS-C subjects.

Reviewer #3: Dear authors,

Thank you for your invitation to review this manuscript. Jørgensen et. al presented an interesting cross-sectional study which investigates the association between various functional somatic disorders (FSDs) and plasma lipid profiles. Compared to previous studies, the present has the advantage of a considerable number of included participants. The topic is important, the manuscript is generally well written and the statistical methodology adequate. One noticeable drawback is the lack of pertinent reporting guidelines. Another issue mentioned as a limitation is the possibility of reverse causation.

I have the following amendments to improve the quality of this study:

- Abstract, please also present the effect sizes

- Results: the authors should also present the results of the linearity assessments described in the statistical analysis section.

- Discussion and conclusion: the authors should elaborate more on the possible explanation of their findings. How could they explain the fact that the association between bodily distress syndrome and lipids was significant, whereas the corresponding association to irritable bowel was not.

6. PLOS authors have the option to publish the peer review history of their article (what does this mean?). If published, this will include your full peer review and any attached files.

Reviewer #1: No

Reviewer #2: No

Reviewer #3: No

---

## [Author Response · Author response to Decision Letter 0]

22 Nov 2023

5. Review Comments to the Author

Reviewer #1: This paper presents the results of primary research which have not, to my knowledge, been published previously. It has a number of strengths including a large, population-based sample and objective outcome measures. The research meets high ethical standards.

Answer: Thank you for these positive comments

There a number of problems with the work and its presentation. These are:

1) Although the sample is truly population-based the low response rate (30%) raises the possibility of a biased sample. Are there other publications from DanFunD that indicate it is representative of the local population?

Answer: Yes, in the discussion (line 355-358) we already stated the following: “Second, the response rate of 29.5% may be considered low, and even though the risk of selection bias is markedly reduced compared to clinical studies, we cannot rule it completely out. But we have shown that selection bias is not a major problem in the DanFunD study (37)”. Furthermore, et should be recognized that exposure-disease associations are only influenced by selection bias, if both exposure and disease are associated with participation.

2) The introduction (line 87) refers to “the lack of large-scale, population-based, epidemiological studies, where the various FSD are analyzed simultaneously (17). This is a misquote. Reference 17 presents the “case for longitudinal population-based studies in the field of functional somatic syndromes.” This is a serious shortcoming of the present study which is cross-sectional, although the authors do admit (line 358) “ we are dealing with a cross-sectional design, and therefore cannot determine whether the findings are consequences or determinants for FSD.

Answer: Thank you for this comment. The reviewer acknowledge that we made it clear that the present manuscript is dealing with a cross-sectional design. We quoted ref 17, as DanFunD is planned as a longitudinal study. But reviewer is right: This reference is irrelevant here and is now deleted.

3) The literature review is rather limited. A recent paper (see A below) found 13 studies that had examined lipid profiles as predictors of FSS. That paper quoted evidence that a raised BMI predicted FM whereas the opposite was true in IBS. Another prospective population-based study also reported this finding (B below).

Answer: Thank you for this comment. We do not refer all articles, as this is not a systematic review, but instead have chosen – in our view – the most relevant articles for citation. Here we focus on articles with the main purpose to look at possible associations between FSD and lipid profile (including metabolic syndrome where blood lipids are central in the definition). The review by Kitselaar is a very comprehensive review of many cohort studies looking at nearly any possible predictors of persistent somatic symptoms. In the review the word hyperlipidemia is only mentioned once under “inconsistent findings”, and a full overview is referred to an on-line appendix. Here we find 14 articles where the word hyperlipidemia is mentioned. Most studies are from the Taiwanese/Chinese National Health Insurance Research Database (NHIRD), which is a register study, where various diagnosis can be linked (e.g. ICD for hyperlipidemia with ICD for any FSD). None of the studies had as a primary aim to investigate any association between lipid metabolism and FSD and details about hyperlipidemia are lacking. The article by Monden et al. looked at close to 40 possible predictors (within a large range of various subjects), where a “lifetime diagnosis of high cholesterol” was associated with development of self-reported CFS. These articles do not – in our opinion - contribute to a deeper understanding of possible association between lipid metabolism and FSD. We have chosen to mention these two references in the discussion part to illustrate the present difficulties within research in FSD. 

The reviewer's focus seems to be on the association between BMI and FSD, but this is not the focus of the present paper. Due to a known association between BMI and FSD and BMI and blood lipids we adjust for BMI in model two in our analyses. In a search for any associations between anthropometry and FSD much more data than BMI should be included, which should be the subject of another manuscript. 

4) The theoretical basis for the whole project is not very clear. The stated aim is “to investigate possible associations between plasma lipid profiles and Functional Somatic disorders in a large population sample”. But the functional somatic disorders represents a group of disorders and there is no theoretical reason why all the different disorders (BDS single, Multi BDS, CF, IB, CWP -pure types and with other syndromes) should all have the same relationship with lipid profiles. Why should single BDS , which may be cardiopulmonary, gastrointestinal, musculoskeletal or general symptoms type share a common relationship with lipid profiles? The theoretical basis should be explained more fully.

Answer: Thank you for this comment. We have now tried to make this more clear. We do not claim that all the different disorders are the same and should have the same relation to lipid metabolism, but frankly we do not know. This is actual one of the crucial questions in this area: Are we dealing with one or several diseases? But very little data are available for supporting the one theory above the other. That is why we are riding two horses in DanFunD looking at both the unifying diagnostic concept (BDS) and the individual FSS’s to see if there indeed would be differences in the lipid profile between them. 

Our aim was thus to explore whether any solid associations between lipid profile and FSD exist. The literature – including the two suggested articles (A and B) by the reviewer – clearly shows that such a study is necessary. As little is known about the biological etiology of FSD, this approach should be the first step. 

Another drawback in the literature is the way the individual FSS’s are handled. E.g., when looking at CFS, CWP/FM, and IBS the literature does not distinguish between the cases with or without co-morbid FSD. This is a serious drawback, as it is well known that a major overlap between these syndromes exist. To understand any associations with biological markers and FSD we need to look at both case with and without co-morbid FSD – otherwise the found associations with e.g. IBS could be explained by a co-morbid FSD such as CSF. 

We have changed the wording in the end of the introduction, which we hope will clarify our intentions.

5) This lack of clarity has led to multiple statistical tests. The figures indicate that 112 tests were performed and there seems to be little theoretical rational that can guide the reader through such a myriad of results. The outcome measure consists of 4 measures (total cholesterol, non-HDL cholesterol, HDL-cholesterol, triglycerides). These 4 measures are compared in each of 5 diagnostic categories with syndrome-free participants but each of the diagnostic categories are further split into pure and other types. This muddies the waters a lot. The results have then been adjusted for BMI, lifestyle, and social factors and some analysis are repeated in the sexes separately and by age groups. A clearer statistical plan could, surely, reduce the number of test results.

Answer: Thank you for these comments, but we respectfully disagree with the reviewer – including the reviewers “no” to “Has the statistical analysis been performed appropriately and rigorously?”. In our opinion we followed a very strict analytic plan. It should be acknowledged, though, that a large number of tests were planned and performed. We believe this relatively explorative approach is warranted as the a priori knowledge of specific associations and underlying biological mechanisms was relatively limited. To explore any associations between lipid metabolism and FSD it is necessary to look more detailed into the lipids, where these are analyzed according to the literature. Just looking at total cholesterol or “hyperlipidemia” is not sufficient. Cholesterol can be divided into high-density-lipoprotein (HDL) cholesterol, low-density-lipoprotein (LDL) cholesterol and very low-density-lipoprotein (VLDL) cholesterol. For practical reasons it is most appropriate to use total cholesterol, HDL (“good”) cholesterol and non-HDL (“bad”) cholesterol. VLDL cholesterol is very much a part of triglycerides. Therefore, a minimum of four measures are needed. Also, we use various delimitations of FSD for which we have argued in the manuscript. This of course led to many tests. We have discussed using a Bonferroni correction, but one of the important presumptions before using this correction is that the analyses should be independent of each other’s, which they obviously are not. As we are dealing with a random sample from the population and not a narrow clinical sample, we must acknowledge that persons with cardio-metabolic disorders could have a lipid profile that could bias the results. Therefore, we have repeated the analyses excluding persons with cardio-metabolic disorders in supplementary tables – now transferred into an appendix.

6) The very large number of tests means that no clear conclusion can be reached. The statement that “an association between some delimitations of FSD and an unfavorable lipid profile” is not satisfactory. In fact, the results are compatible with interpretation that BDS is a group of different disorders that have little in common. Not only do the characteristic symptoms differ in CF, IB and CWP, they appear to have a different relationship with lipid levels. The most likely explanation is that some of the subgroups of CFS have a raised BMI which may concur with the present results (D). Overall the results suggest that the notion of lumping together different disorders into a single group makes little sense when it comes to assessing correlates. It is known that BMI does not predict numerous somatic symptoms (the hallmark of Multi BDS) see C. A Kitselaar M, van der Vaart R et al. Predictors of Persistent Somatic Symptoms in the General Population: A Systematic Review of Cohort Studies Psychosomatic Medicine 2023 Jan 1;85(1):71-78.

B Monden, R, Rosmalen, JGM et al Predictors of new onsets of irritable bowel syndrome, chronic fatigue syndrome and fibromyalgia: the lifelines study. Psychol Med. 2022 Jan;52(1):112-120.

C Creed F. The Predictors of Somatic Symptoms in a Population Sample: The Lifelines Cohort Study. Psychosom Med 2022 Nov-Dec;84(9):1056-1066.

D Vollmer-Conna U, Aslakson E, White PD. An empirical delineation of the heterogeneity of chronic unexplained fatigue in women. Pharmacogenomics 2006 Apr;7(3):355-64.

Answer: Reviewer suggests that the results are compatible with the interpretation that BDS is a group of different disorders that have little in common. We do touch upon this in the beginning of our discussion section, where we conclude that IB is somewhat different from CWP, CF and BDS. We are aware of the dispute between “splitters” and “lumpers”, and we find it a very intriguing discussion. But we think that we are not there yet that such a firm conclusion can be made. We should wait with firm conclusion until more data are available. We have further elaborated on this in the discussion. That numerous somatic symptoms are the hallmark of Multi BDS is a misunderstanding. It is mainly defined based on characteristic symptom pattern and not on the number of symptoms.

The reviewer speculates that the most likely explanation is that some of the subgroups of CFS have increased BMI which may concur with the present results. The reviewer cites a smaller case-control study of the highly selected material using latent class analyses. We agree that BMI can play a role, and in our analyses, we included BMI as a confounder, which did not change the results substantially. 

Minor points

7) Abstract Line 44. This should read .”.. validated self – administered questionnaires if this was the case.

Answer: Thank you – this has now been changed

8) Spelling mistakes eg. (lines 186, 190) Tabel 1 and 2 In table 2 “To few cases” (should be Too few..)

Answer: Thank you – this has now been changed

9) Figures add little because so many similar boxes make it impossible to spot differences where these occur.

Answer: Thank you for this comment. The figures are not crucial for the conclusions, but they more clearly show the distribution of the lipids in the various delimitations and underline the small differences. So, if the Editor accept these figures, we would like to keep them. 

Reviewer #2: The manuscript is very interesting, addressing a topic of interest, while it is well-written and structured.

Answer: Thank you very much for these positive comments

Some minor comments:

1. I think that the definition of cardio-metabolic diseases should also include MAFLD, CKD, etc., besides atherosclerotic CVD and diabetes mellitus.

Answer: Thank you. It is aways a big discussion which diseases should be excluded. We prefer not to exclude too many, but we have – in sensitivity analyses – excluded the major relevant diseases on which we have information. Unfortunately, we do not have information on MAFLD and CKD. But it is noteworthy how little the results change.

2. Information regarding the type of lipid-lowering treatment (statin vs. non-statin) would be useful.

Answer: Thank you for this comment - we have now added that we are talking about statin use.

3. Of note, did the researchers have any available data concerning the association of IBS with lipid parameters, according to type of IBS (IBS-D, IBS-C, IBS-M), since, for example, subjects with IBS-D may experience more strict dietary pattern than IBS-C subjects.

Answer: Unfortunately, we do not have this information in the cohort

Reviewer #3: Dear authors,

Thank you for your invitation to review this manuscript. Jørgensen et. al presented an interesting cross-sectional study which investigates the association between various functional somatic disorders (FSDs) and plasma lipid profiles. Compared to previous studies, the present has the advantage of a considerable number of included participants. The topic is important, the manuscript is generally well written and the statistical methodology adequate. One noticeable drawback is the lack of pertinent reporting guidelines. Another issue mentioned as a limitation is the possibility of reverse causation.

Answer: Thank you for these positive comments. We are not quite sure what the reviewer means with “the lack of pertinent reporting guidelines”? Reverse causation is of course always a problem in most cross-sectional studies. That’s why we stress that we cannot say whether the associations found could be either a cause or a consequence.

I have the following amendments to improve the quality of this study:

- Abstract, please also present the effect sizes

Answer: Thank you for this suggestion. We think that this suggestion would increase the word count in the abstract considerable (and thereby exceed the recommended word count) and make the abstract difficult to read. 

- Results: the authors should also present the results of the linearity assessments described in the statistical analysis section.

Answer: This is possible, but it is rather voluminous and not normally done. We can provide an appendix if the Editor request this.

- Discussion and conclusion: the authors should elaborate more on the possible explanation of their findings. How could they explain the fact that the association between bodily distress syndrome and lipids was significant, whereas the corresponding association to irritable bowel was not.

Answer: Thank you for this comment. Please see our comments to reviewer 1 and the amendment done in the text.

6. PLOS authors have the option to publish the peer review history of their article (what does this mean?). If published, this will include your full peer review and any attached files.

Do you want your identity to be public for this peer review? For information about this choice, including consent withdrawal, please see our Privacy Policy.

Reviewer #1: No

Reviewer #2: No

Reviewer #3: No

---

## [Decision Letter · Decision Letter 1]

19 Dec 2023

Lipid metabolism and functional somatic disorders in the general population. The DanFunD study

PONE-D-23-13283R1

Dear Dr. Jørgensen,

We’re pleased to inform you that your manuscript has been judged scientifically suitable for publication and will be formally accepted for publication once it meets all outstanding technical requirements.

Kind regards,

Aleksandra Klisic

Academic Editor

PLOS ONE

---

## [Editor Report · Acceptance letter]

16 Jan 2024

PONE-D-23-13283R1 

PLOS ONE

Dear Dr. Jørgensen, 

I'm pleased to inform you that your manuscript has been deemed suitable for publication in PLOS ONE. Congratulations! Your manuscript is now being handed over to our production team.

Kind regards, 

on behalf of

Dr. Aleksandra Klisic 

Academic Editor

PLOS ONE